# Antifungal Activity of Ginger Rhizome Extract against *Fusarium solani*

**Ke-Yong Xi** [1,†], **Shi-Jie Xiong** [1,†], **Gang Li** [1], **Chang-Quan Guo** [1], **Jie Zhou** [1], **Jia-Wei Ma** [1,2], **Jun-Liang Yin** [3,*], **Yi-Qing Liu** [1,*] and **Yong-Xing Zhu** [1,*]

1   Spice Crops Research Institute, College of Horticulture and Gardening, Yangtze University, Jingzhou 434025, China
2   Jingzhou Jiazhiyuan Biotechnology Co., Ltd., Jingzhou 434025, China
3   College of Agriculture, Yangtze University, Jingzhou 434025, China
*   Correspondence: yinjunliang@yangtzeu.edu.cn (J.-L.Y.); liung906@163.com (Y.-Q.L.); xbnlzyx@163.com (Y.-X.Z.)
†   These authors contributed equally to this work.

**Abstract:** *Fusarium solani* is one of the most ruinous soil-borne pathogens and seriously affects crop yields and quality worldwide. Ginger is an important medicinal crop, and ginger rhizome extract (GRE) has been used as an alternative for chemical fungicides and bactericides. We investigated the fungistatic effect of GRE on *F. solani* and analyzed the chemical constituents of GRE using UPLC-MS/MS. Antifungal assay results showed that 20 mg/mL of GRE completely inhibited the growth of *F. solani*. Morphological analysis revealed that GRE destroyed the morphology and structure of mycelia, thus inhibiting mycelial growth. Furthermore, GRE suppressed the activities of cell wall-degrading and cellular respiratory-related enzymes and decreased the content of fusaric acid, which reduced or even abrogated the infection ability of *F. solani*. UPLC-MS/MS analysis showed that GRE constituents belonged to eight categories, among which phenolic acids were the highest in content (46.29%) and tannins were the lowest in content (0.06%). When the antifungal activities of major phenolic and flavonoid compounds were evaluated, 4.0 mg/mL 4-hydroxybenzaldehyde and 15.0 mg/mL quercetin were found to completely inhibit *F. solani* growth. These results highlight GRE as an excellent source of antifungal compounds and suggest the possibility of using 4-hydroxybenzaldehyde and quercetin as natural fungicides to control crop diseases.

**Keywords:** antifungal activity; ginger rhizome extract; *F. solani*; fusarium acid

## 1. Introduction

Plant pathogens can cause serious harm to crops in the production and postharvest stages, reducing global yields by up to 30% and resulting in low-quality grains, fruits, and vegetables. Among crops, the growth of crops is seriously threatened by abiotic stresses and biotic stresses [1]. Fusarium wilt, caused by *Fusarium oxysporum* and *Fusarium solani*, poses significant threats to production and post-harvest storage of many fruits and vegetables [2–4]. Fusarium spreads via the soil, wind, and irrigation water, and generally invades plant tissues via wounds. It causes huge economic losses because of rotting and softening of the stem in many fruits and vegetables including banana [5], tomato [6], pigeon pea, and even almost any plant species [7].

Currently, chemical fungicides are mainly used to control fusarium wilt; however, due to anthropogenic activities including improper application of fertilizers [8], the intense use of chemical fungicides has caused many serious problems, such as environmental pollution, human and animal health concerns, and ecological imbalances. And the frequent use of fungicides, pathogens will develop resistance under the selection of chemical fungicides and show decreased sensitivity to chemical fungicides [9]. Therefore, natural fungicides with high efficiency, low toxicity, and no residues need to be further explored. Studies

have shown that numerous plant extracts have inhibitory effects on fungal diseases, and studies on botanical fungicides and antifungal plant substances have attracted research attention [10].

Ginger (*Zingiber officinale*) has been used as a spice and medicine for over 200 years in Traditional Chinese Medicine. The obtained findings suggest potential of ginger extract as an additive in the food and pharmaceutical industries [11,12]. Ginger contains various ingredients with antifungal activity and of medicinal value. Results indicate that ginger contains monoterpenoids, sesquiterpenoids, phenolic compounds, and its derivatives, aldehydes, ketones, alcohols, esters, which provide a broad antimicrobial spectrum against different microorganisms and make it an interesting alternative to synthetic antimicrobials [13,14]. For example, 6-gingerol inhibits the proliferation of human cervical cancer cells and induces their apoptosis [15]. Recent studies have shown that both ginger extract and ginger essential oil have antifungal activities against plant pathogens, such as *Fusarium oxysporum* and *Colletotrichum falcatum* [16,17]. Noshirvani et al. reported that ginger essential oils could be used to plasticize chitosan-carboxymethyl cellulose films while improving moisture permeability and maintaining antifungal activity. Similarly, Agarwal et al. proposed that 6-dehydroshogaol isolated from ginger rhizomes exhibited maximum insect growth regulatory activity while dehydrozingerone imparted maximum antifungal activity. These findings indicate that ginger extract might be useful as an alternative for fungicides and bactericides [18,19].

In a preliminary study, we found that when ginger plants were inoculated with *F. solani*, the stem base rotted and the leaves turned yellow and wilted within one week, whereas the underground rhizome showed no or lighter symptoms, which indicated that ginger rhizome may contain antifungal compounds that exert inhibitory effects on *F. solani*. Moreover, we evaluated the *F. solani* resistance of five main varieties in China and found that 'Shandongdajiang' is the most resistance cultivar. Therefore, in this study, we explored the antifungal effects of GRE extracted from 'Shandongdajiang' and determined the antifungal chemical composition of GRE. We hoped this study would provide a theoretical basis for research on and utilization of GRE as a plant-derived fungicide for controlling fusarium wilt.

## 2. Materials and Methods

### 2.1. Plant Materials

The plant materials were collected in the Agricultural Science and Technology Industrial Park of Yangtze University (Jingzhou, Hubei Province, China, 112.067107° E, 30.369843° N). Healthy ginger plants (cv. Shandongdajiang) were selected for experiments after 180 days of growth, and healthy ginger rhizomes with no mechanical damage were harvested 180 days after planting, transported to the laboratory, and used for experiments within 2 h.

### 2.2. Preparation of Fungal Pathogen

*F. solani* was cultured on potato dextrose agar (PDA) containing 2% dextrose ($w/v$) and 1.5% agar ($w/v$) at 28 °C in the dark for 7 days. The conidia were washed off from the PDA plates with distilled water. The spore suspension was filtered through two layers of sterile gauze to remove adherent mycelia. The concentration of the spore suspension was adjusted to $1 \times 10^7$ spores/mL after counting the spores using a hemocytometer [20].

### 2.3. Plant Extraction

GRE was prepared according to the method reported by Tanweer et al. [21]. About 50 kg healthy ginger rhizomes were washed with distilled water, cut into pieces, and dried in an oven at 60 °C for 72 h. The dried rhizome pieces were grinded into a powder using a blender and filtered through a 100-mesh sieve. The filtered powder (5 g) was dissolved in 70% ethanol (5 mL) and extracted in a Soxhlet extractor (SXT-06, Shanghai Benang Scientific Instruments Co., Ltd., Shanghai, China) at 80 °C for 9 h. All extracts were

collected, and the alcohol was evaporated using a rotary evaporator (R-1001VN, Shenzhen Kemis Technology Co., Ltd., Shenzhen, China) at 60 °C for 1 h to obtain a paste. One hundred milligrams of the paste was dissolved in 1 mL of 30% ethanol and filtered through a 100-mesh sieve; the filtered liquor was the GRE. The GRE was stored at 4 °C until use.

*2.4. Antifungal Activity Assays*

2.4.1. Effect of GRE on *F. solani* Growth

Determination of Mycelial Growth

To determine the inhibitory effect of GRE on mycelial growth of *F. solani*, different concentrations of GRE (0, 1.25, 2.5, 5, 10, and 20 mg/mL) or 30% ethanol as a control were added to melted PDA, which was poured into 60-mm-diameter petri dishes [22]. Six-millimeter-diameter mycelial disks taken from 7-day-old PDA cultures were placed in the center of the petri dishes, which were then incubated at 28°C in the dark. Five days later, the dishes were photographed to assess the colony growth status and measure the colony diameter (cm). Mycelia were frozen in liquid nitrogen and then stored at −80 °C for analysis. The assay was performed in three replicates.

Determination of Spore Germination

The spore germination assay was performed referring to Yun et al. [23]. Briefly, 1 mL of spore suspension ($1 \times 10^7$ spores/mL) was added into 60-mm-diameter petri dishes containing 5 mL of PDA with different concentrations of GRE (0, 1.25, 2.5, 5, 10, and 20 mg/mL) or 30% ethanol. The dishes were incubated at 28 °C in the dark. Six hours later, approximately 200 spores were randomly selected to calculate the germination rate using a microscope. Each treatment included three replicates.

$$\text{The spore germination rate (\%)} = (\text{Number of germinated spores}/\text{Total number of spores}) \times 100 \qquad (1)$$

Morphological Observation of Fungal Hyphae

Scanning electron microscopy (SEM, Ion Sputter JFC-1100, Tokyo, Japan) was used to observe morphological changes in *F. solani* hyphae treated with 10 mg/mL of GRE according to a previously published method with some modifications [24]. In brief, the 6 mm-diameter mycelial disks taken from 7 days' culture of pathogen were placed in the center of each petri dish containing 5 mL PDA with 10 mg/mL of GRE, and the PDA with 30% ethanol was used as control. Then, a cover slip (9 cm × 1.5 cm) was inserted into the center of petri dish at 45° in sterile condition. After culturing for 5 days, the cover slip was gently pulled out and the excess medium on the cover slip was washed off with phosphate buffer, then the cover slip was soaked in 2.5% glutaraldehyde solution for 12 h at 4 °C. The mycelium on the cover slips was washed three times with sterile distilled water for 5 min, then serially dehydrated with 30%, 50%, 70%, 90%, 100% of alcohol and tertiary-butyl alcohol solution, respectively, every 10 min. Finally, the samples were dried by a vacuum freeze dryer, then sprayed gold for 45 s with an ion sputter coater, and the mycelium from 1–2 mm of the tip were observed by SEM. The number of measurements was 5 per treatment and each measurement contained 10–20 field of microscope.

2.4.2. Effect of GRE on the Cell Membrane Integrity of *F. solani*

Determination of the Relative Conductivity

The relative conductivity of *F. solani* was measured according to Lee et al. [25]. One milliliter of a spore suspension ($1 \times 10^7$ spores/mL) was added into 100 mL of potato dextrose broth (PDB). The cultures were incubated in an incubator shaker at 28 °C, 180 rpm in the dark for 2 days. Then, the mycelia were collected and washed with sterile distilled water. The mycelia (2.5 g, wet weight) were suspended in 25 mL of sterile distilled water containing different concentrations of GRE (0, 1.25, 2.5, 5, and 10 mg/mL) and incubated under shaking at 180 rpm at 28 °C in the dark. The conductivity of *F. solani* was measured using a conductivity meter after 1, 2, 3, 4, 5, and 6 h of treatment with GRE and recorded

as $C_1$. The conductivity of *F. solani* after 0 h of treatment with GRE was measured and recorded as $C_0$. After 6 h of treatment, the samples were heated in boiling water for 10 min, and the conductivity was measured and recorded as $C_2$. Three replicates were performed for each treatment.

$$\text{The relative conductivity (\%)} = (C_1 - C_0)/(C_2 - C_0) \times 100. \tag{2}$$

Determination of the Soluble Protein Content

The protein content of mycelia was determined according to the Bradford method [26]. As described above, mycelia were harvested after culturing on PDA with 10 mg/mL of GRE or 30% ethanol (control). The mycelia (0.5 g) were grinded into a homogenate in phosphate buffer and the homogenate was centrifuged at $4000\times g$ at 4 °C for 10 min. Then, 0.2 mL of supernatant was mixed with 0.8 mL of distilled water and 5 mL of Coomassie brilliant blue G-250 (Solarbio, Beijing, China). The absorbance at 595 nm was measured using a UV spectrophotometer (UV-8000S, Shanghai Metash Co. Ltd., Shanghai, China).

Determination of the Soluble Sugar Content

Mycelia (0.02 g) were added into 1 mL of distilled water, boiled for 30 min, and centrifuged at $4000\times g$ for 10 min. The soluble sugar content in the supernatant was quantified using the anthrone sulfuric acid assay [27]. Briefly, the supernatant was diluted to 25 mL with distilled water, which was the solution to be tested. Then, 0.2 mL of the solution was mixed with 1.8 mL of distilled water, 0.5 mL of ethyl anthrone acetate, and 5 mL of sulfuric acid, shaken, and then held in a boiling water bath for 1 min. Let it cool to room temperature, then the absorbance value was measured at 630 nm, and a standard curve was drawn based on the absorbance in different concentrations (μg/mL) of dextrose solution.

2.4.3. Effect of GRE on the Cell Membrane Integrity of *F. solani*
Enzyme Extraction

One milliliter of a spore suspension ($1 \times 10^7$ spores/mL) was added into 100 mL of PDB containing 10 mg/mL GRE or 30% ethanol (control). On days 1, 2, 3, 4, and 5, the mycelia were washed off using distilled water and collected. Approximately 0.5 g mycelia were grinded in 1 mL of phosphate buffer (0.1 M, pH 7.5) and the mixture was subjected to ultrasonic cell crushing for 5 min (200 W, 2-s intervals) to break the cells. After centrifugation at $4000\times g$ at 4 °C for 10 min, the supernatant was collected and used as crude enzyme solution.

Determination of Pectinase Activity

Crude enzyme solution (0.25 mL) was added into 0.75 mL of 2% pectin solution and incubated at 50 °C for 30 min. Then, 0.75 mL of dinitrosalicylic acid reagent [28] was immediately added. In control tubes, 0.75 mL of distilled water was added instead of 2% pectin solution. The reaction mixtures were soaked in boiling water for 10 min, cooled, and adjusted to 10 mL with distilled water. The absorbance at 540 nm was measured using a UV spectrophotometer, and a standard curve was drawn based on the absorbance in different concentrations (μg/mL) of d-galacturonic acid [29].

Determination of β-Glucosidase Activity

Crude enzyme solution (50 μL) was mixed with 250 μL of sodium acetate buffer (0.1 M, pH 4.5) and 250 μL of p-nitrophenyl β-d-glucopyranoside (4 mM, Sigma) and the mixture was incubated at 50 °C for 10 min. Then, 2 mL of sodium carbonate (2 M) was added to stop the enzymatic reaction, and the absorbance of the released product at 410 nm was determined using a spectrophotometer. One unit of enzyme activity was defined as the amount of enzyme required to release 1 μmol of nitrophenol per minute of reaction [30].

2.4.4. Determination of Respiratory Metabolic Pathway Enzyme Activities

The activities of MDH and SDH were determined using malate and succinate dehydrogenase assay kits (Nanjing Jiancheng Bioengineering Institute, Nanjing, China), respectively.

2.4.5. Determination of Fusaric Acid (FA) Content

The FA content was measured according to Bacon et al. [31] with some modifications. The standard fusaric acid (Sigma Co. Ltd., Creamridge, NJ, USA) was dissolved in acetone with the concentrations of 0, 2, 4, 6, 8, 8, 10, 12 mmg/mL, respectively, then the absorbance of which were determined at 268 nm and the standard curve was drawn. Then, 1 mL spore suspension ($1 \times 10^7$/mL) was added to 100 mL PDB liquid culture medium containing 30% ethanol (control) or 10 mg/mL GRE and cultured with the temperature of 28 °C at 180 rpm for 5 days. Five days later, the mycelia were removed by vacuum filtration and the culture medium was obtained. The equal volume of ethyl acetate was added to the culture medium, then the mixture was extracted three times with a separating funnel and dried to obtain crystal in the fume cupboard. The crystal was dissolved in 5 mL of anhydrous ethanol and absorbance was measured at 268 nm by microplate spectrophotometer (WX-RPC2, WanxinSoft, Shanghai, China), the content of FA was calculated according to the standard curve.

*2.5. Compositional Analysis of GRE by Ultra-Performance Liquid Chromatography-Tandem Mass Spectrometry (UPLC-MS/MS)*

The chemical components of GRE were analyzed by ultra-performance liquid chromatography (UPLC, SHIMADZU Nexera X2, Kyoto, Japan) and tandem mass spectrometry (MS/MS, Applied Biosystems 4500 QTRAP, Waltham, MA, USA). The chromatographic column of UPLC was Agilent SB-C18 (2.1 mm × 100 mm, 1.8 μm). The mobile phases used for gradient elution were: phase A: 0.1% formic acid in ultrapure water; phase B: 0.1% formic acid in acetonitrile. The mobile phases gradient was set as follows: Linearly increased from 5% to 95% B from 0 to 9 min and kept at 95% for one minute; and then decreased to 5% B from 10 to 11 min and kept at 5% from 11 to 14 min. The flow rate: 0.35 mL/min; column temperature: 40 °C; injection volume: 4 μL. LIT and triple quadrupole (QQQ) scanning was performed on a triple quadrupole linear ion TRAP mass spectrometer (Q TRAP, AB4500 Q TRAP UPLC/MS/MS). The system is equipped with ESI turbo ion spray interface and can be controlled by Analyst 1.6.3 software (ABSciex) to run positive and negative ion modes. ESI source operation parameters are as follows: ion source, turbo spray; the source temperature was 550 °C; ion spray voltage (IS) 5500 V (positive ion mode)/−4500 V (negative ion mode); ion source gas I (GSI), gas II (GSII), and curtain gas (CUR) was set as 50, 60, and 25 psi, respectively, and the collision-induced ionization parameter was set as high. Instrument tuning and mass calibration were performed with 10 and 100 umol/L polypropylene glycol solutions in QQQ and LIT modes, respectively. QQQ scanning uses MRM mode with collision gas (nitrogen) set to medium. The DP and CE of each MRM ion pair were completed with further optimization. Specific set of MRM ion pairs was monitored in each period based on the metabolites eluted within each period.

*2.6. Determination of the Effects of 4-Hydroxybenzaldehyde and Quercetin on F. solani Growth*

4-Hydroxybenzaldehyde and quercetin (Macklin, Shanghai, China) were dissolved in 30% ethanol at 100 mg/mL. Six-millimeter-diameter mycelial disks taken from 7-day-old PDA cultures were placed in the center of petri dishes containing 5 mL of PDA with different concentrations of 4-hydroxybenzaldehyde (0, 2.0, 2.5, 2.75, 3.0, and 4.0 mg/mL), quercetin (0, 1.0, 4.0, 6.0, 10.0, and 15.0 mg/mL), or 30% ethanol, and incubated at 28 °C in the dark. Five days later, the dishes were photographed to assess the colony growth status and measure the colony diameter (cm). Three replicates were performed for each treatment.

*2.7. Statistical Analysis*

Descriptive statistics including the mean and SE, along with the Tukey range test for multiple comparison procedure were used when the ANOVA was significant ($p < 0.05$).

Data were first tested for normality with the Kolmogorov–Smirnov test and for homogeneity of variance with the Brown–Forsythe test. SigmaPlot v14.0 (Systat Software Inc., San Jose, CA, USA) was used for statistical analyses and picture drawing.

## 3. Results

### 3.1. Effect of GRE on F. solani Growth

#### 3.1.1. Effects of GRE on Mycelial Growth and Spore Germination

As shown in Figure 1A, the colony diameter of *F. solani* decreased with increasing GRE concentration after culturing for 5 days. Compared with the control treatment, 1.25, 2.5, 5, 10, and 20 mg/mL GRE significantly decreased the colony diameter by 21.44%, 55.02%, 76.62%, 89.27%, and 100%, respectively. Specifically, 20 mg/mL GRE completely inhibited the mycelial growth of *F. solani*. To eliminate the effect of ethanol, the effect of 30% ethanol on *F. solani* colony growth was investigated. The results indicated that 30% ethanol hardly affected colony growth ($p > 0.05$; Figure 1B).

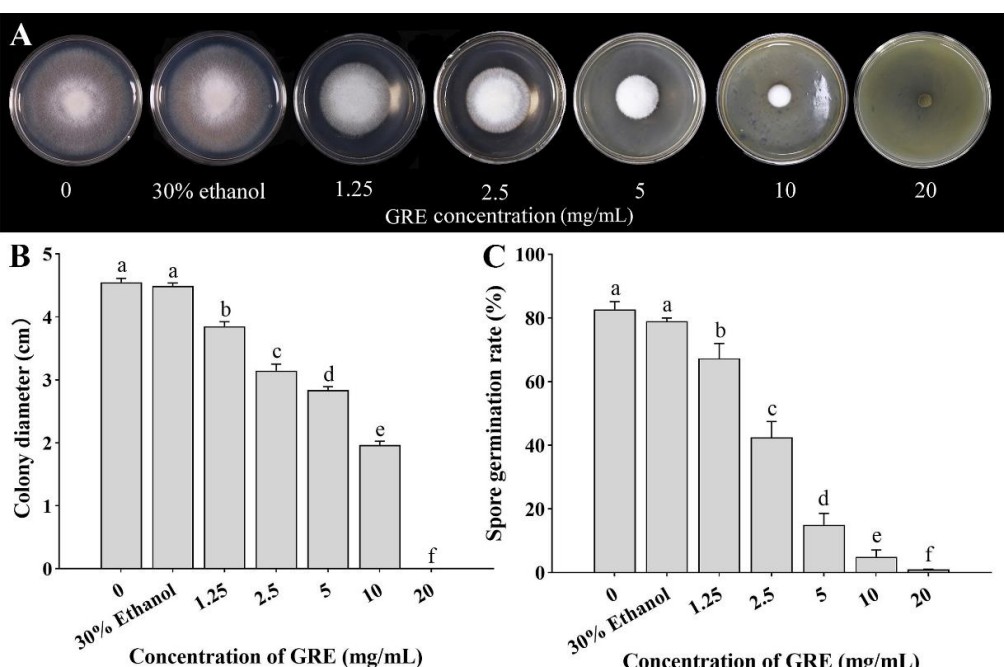

**Figure 1.** Effects of different concentrations of GRE on *F. solani* growth. (**A**) Colony morphology after 5 days of treatment. (**B**) Colony diameter after 5 days of treatment. (**C**) Spore germination rate after 6 h of treatment. The bars on the columns represent standard deviations, and different letters above the bars represent significant differences ($p < 0.05$).

The spore germination rate gradually decreased with increasing GRE concentration after 6 h of treatment. Compared with the control treatment and 0 mg/mL GRE, 1.25, 2.5, 5, 10, and 20 mg/mL GRE significantly decreased the spore germination rate by 18.29%, 48.78%, 82.93%, 93.9%, and 99.8%, respectively. Again, 30% ethanol had no significant effect (Figure 1C). These results indicated that GRE dose-dependently inhibits *F. solani* growth, with the strongest effect noted at 20 mg/mL.

#### 3.1.2. Effect of GRE on Mycelial Morphology

To investigate the effect of GRE on *F. solani* growth further, SEM was employed to observe morphological changes in fungal mycelia after treatment with 10 mg/mL GRE for 5 days. The results showed that 10 mg/mL GRE affected the morphology of *F. solani* mycelia. Control mycelia were plump and smooth and had a complete structure and smooth surface (Figure 2A). After treatment with 10 mg/mL GRE, the hyphae lost their smoothness and became distorted, deformed, and even fractured (Figure 2B). These results

suggested that GRE affects the morphology and destroys the structure of *F. solani* mycelia, which most likely is responsible for the inhibited mycelial growth.

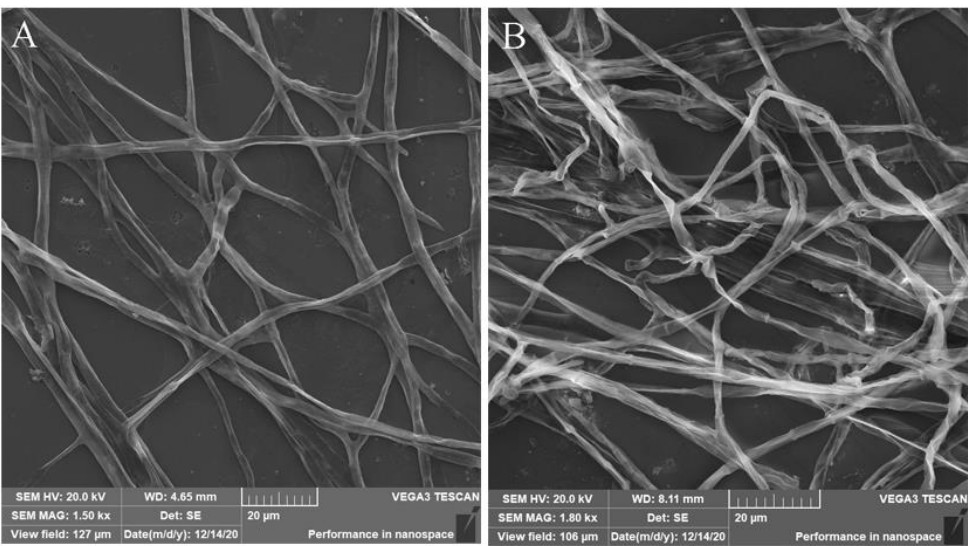

**Figure 2.** SEM images of *F. solani* mycelia. (**A**) Control, treatment with 30% ethanol. (**B**) Treatment with 10 mg/mL GRE.

### 3.2. Effect of GRE on the Cell Membrane Integrity of F. solani

#### 3.2.1. Effect of GRE on the Relative Conductivity of *F. solani*

To evaluate the effect of GRE on the cell membrane permeability of *F. solani*, the relative conductivity was measured. The relative conductivity consistently increased with increasing GRE concentration between 1 to 6 h after treatment. After 6 h of treatment, the relative conductivity of *F. solani* treated with 1.25, 2.5, 5, and 10 mg/mL GRE significantly increased by 1.68-, 3.28-, 5.21-, and 5.95-fold, respectively, as compared to that of the control ($p < 0.05$; Figure 3A). These results indicated that the GRE destroys the *F. solani* cell membrane, leading to leakage of intracellular electrolytes and increased relative conductivity.

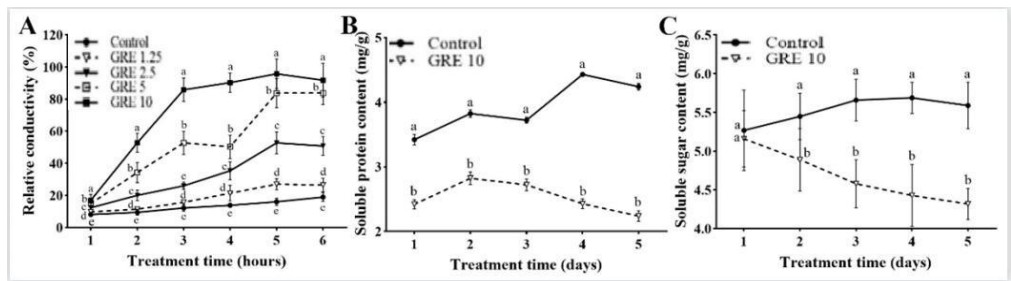

**Figure 3.** Effects of different concentrations of GRE on relative conductivity (**A**) and the effect of 10 mg/mL of GRE on soluble protein (**B**) and soluble sugar (**C**) contents in *F. solani*. The bars on the curves represent standard deviations, and different letters at the same time point represent significant differences ($p < 0.05$).

#### 3.2.2. Effect of GRE on the Soluble Protein Content in *F. solani*

As 20 mg/mL of GRE completely inhibited the growth of *F. solani*, 10 mg/mL of GRE was selected as an optimal concentration in subsequent experiments. Compared with the control treatment, 10 mg/mL GRE significantly decreased the soluble protein content from day 1 to 5. On the day 5, 10 mg/mL GRE decreased the soluble protein content to the lowest value, with a significant decrease of 52.96% when compared with the control (Figure 3B). The soluble protein content in control plants peaked on day 4 and then slightly decreased.

### 3.2.3. Effect of GRE on the Soluble Sugar Content in *F. solani*

As shown in Figure 3C, 10 mg/mL GRE had no obvious effect on the soluble sugar content when compared with the control treatment on day 1, but it significantly decreased the soluble sugar content when compared to the control from days 2 to 5. On day 5, the soluble sugar content reached a minimum, with a 22.72% decrease when compared with the control.

### 3.3. Effect of GRE on Cell Wall Degrading Enzyme Activities in *F. solani*
### 3.3.1. Effect of GRE on Pectinase Activity in *F. solani*

After 10 mg/mL GRE treatment, the pectinase activity in *F. solani* first increased and then decreased, to reach a maximum on day 4. The pectinase activity after 10 mg/mL GRE treatment was consistently significantly lower than that in control from day 1 to 5 (Figure 4A). Compared with the control, 10 mg/mL GRE decreased the pectinase activity by 40.66%, 41.03%, 42.22%, 31.28%, and 38.91% from day 1 to 5, respectively.

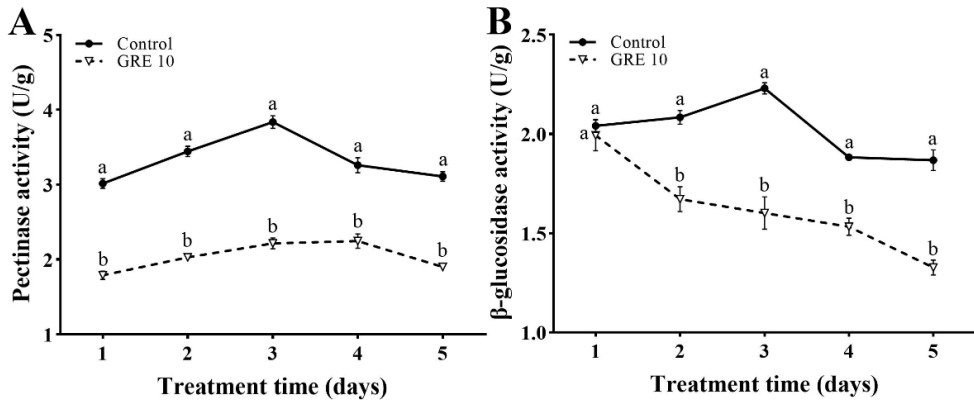

**Figure 4.** Effects of 10 mg/mL GRE on pectinase activity (**A**) and β-glucosidase activity (**B**) in *F. solani*. The bars on the curves represent standard deviations, and different letters at the same time point represent significant differences ($p < 0.05$).

### 3.3.2. Effect of GRE on β-Glucosidase Activity in *F. solani*

The β-glucosidase activity in *F. solani* was decreased significantly after treatment with 10 mg/mL GRE. As shown in Figure 4B, 10 mg/mL GRE had a significant inhibitory effect on the β-glucosidase activity in *F. solani*; it decreased the β-glucosidase activity by 2.4%, 22.5%, 28.25%, 18.62%, and 29.03% from day 1 to 5, respectively.

### 3.4. Effect of GRE on the Respiratory Metabolism of *F. solani*

Treatment with 10 mg/mL GRE gradually suppressed the malate dehydrogenase (MDH) activity from 50% on day 1 to 78% on day 5, respectively, when compared with the control (Figure 5A).

As shown in Figure 5B, 10 mg/mL GRE suppressed the succinate dehydrogenase (SDH) activity. The SDH activity after GRE treatment was higher than that in the control treatment on day 1, but decreased by 11.14%, 39.85%, 55.61%, and 50.17% from day 2 to 5, respectively.

### 3.5. Effect of GRE on the FA Content in *F. solani*

As shown in Figure 6, GRE significantly inhibited FA production when compared with the control treatment. The FA content after 10 mg/mL GRE treatment decreased from day 1 to 5, and it was consistently lower than that in control plants. On day 1, 10 mg/mL GRE did not significantly affect the FA content; however, from day 2 to 5, the FA contents decreased by 13.93%, 29.08%, 46.17%, and 49.56%, respectively, with the strongest effect on day 5.

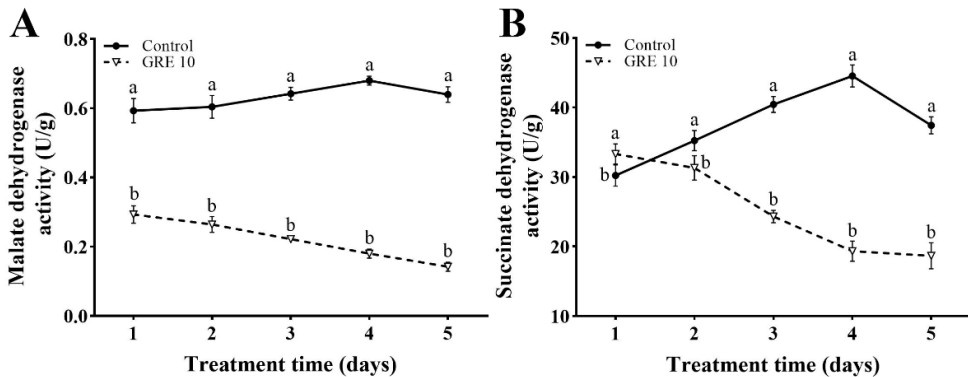

**Figure 5.** Effects of 10 mg/mL GRE on MDH (**A**) and SDH (**B**) activities in *F. solani*. The bars on the curves represent standard deviations, and different letters at the same time point represent significant differences ($p < 0.05$).

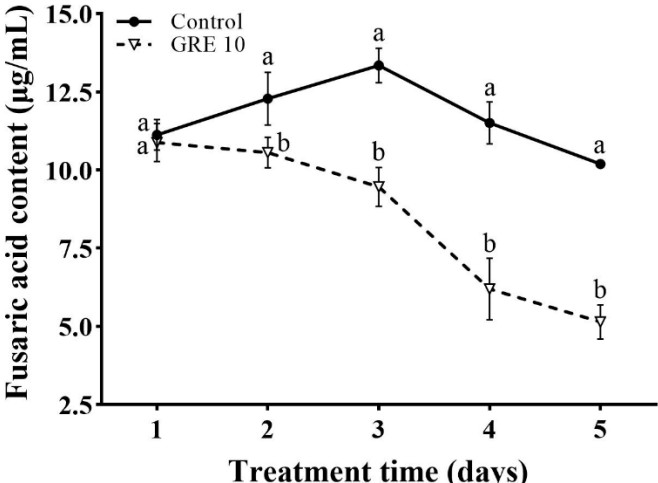

**Figure 6.** Effect of 10 mg/mL GRE on the FA content in *F. solani*. The bars on the curves represent standard deviations, and different letters at the same time point represent significant differences ($p < 0.05$).

### 3.6. Chemical Components of GRE

Using UPLC-MS/MS, 393 components were detected in GRE, which could be divided into eight categories: 147 components were phenolic acids (46.29%), 84 were flavonoids (3.93%), 2 were quinones (3.93%), 30 were lignans and coumarins (2.09%), 7 were tannins (0.06%), 43 were alkaloids (17.96%), 22 were terpenoids (5.18%), and 58 were others (20.55%). Phenolic acids were the most abundant, tannins the least (Table S1).

### 3.7. Effects of 4-Hydroxybenzaldehyde and Quercetin on F. solani

To further explore the antifungal components of GRE, two substances with the following characteristics: (1) reported to possess antimicrobial effects and (2) significant difference in content between rhizomes and aboveground parts (Figure S1) were selected for antifungal assays. The contents of 4-hydroxybenzaldehyde in the rhizome and aboveground parts were significantly different; 4.19% in the rhizome and 0.76% in the aboveground parts. Similarly, the content of quercetin in the rhizome was 0.086% and that in the aboveground parts nearly 0% (Figure S1).

The colony diameter of *F. solani* decreased with increasing concentrations of 4-hydroxybenzaldehyde after culturing for 5 days (Figure 7A). Compared with the control, the colony diameters after 2.0, 2.5, 2.75, 3.0, and 4.0 mg/mL 4-hydroxybenzaldehyde treatments were significantly decreased by 11.56%, 28.57%, 46.26%, 72.11%, and 100%, respectively. Thus, at 4.0 mg/mL, 4-hydroxybenzaldehyde completely inhibited the mycelial

growth of *F. solani* (Figure 7C). Similarly, the colony diameter decreased with increasing concentrations of quercetin after culturing for 5 days (Figure 7B). The colony diameters after 1.0, 4.0, 6.0, 10.0, and 15.0 mg/mL quercetin treatments were significantly decreased by 17.93%, 39.31%, 52.41%, 65.52%, and 100%, respectively (Figure 7D).

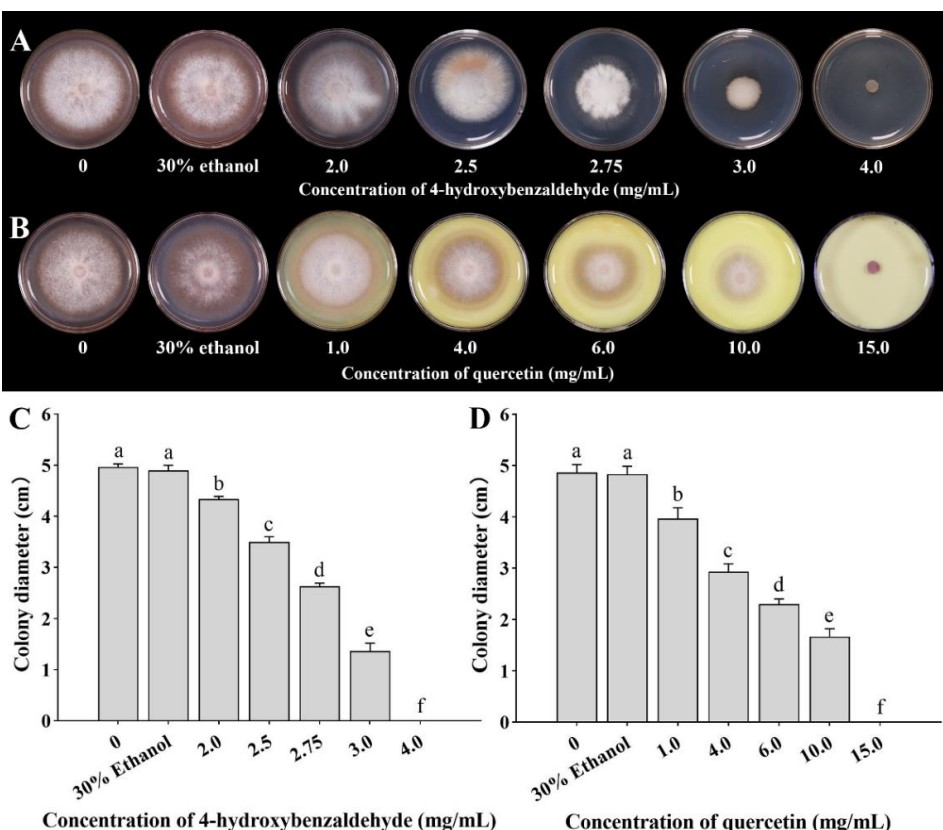

**Figure 7.** Effects of different concentrations of 4-hydroxybenzaldehyde (**A**,**C**) and quercetin (**B**,**D**) on colonial morphology (**A**,**B**) and colony diameter (**C**,**D**). The bars on the columns represent standard deviations, and different letters on the bars represent significant differences ($p < 0.05$).

## 4. Discussion

Numerous studies have shown that ginger extract has a wide range of antimicrobial activities and acts as a botanical fungicide by inhibiting the spore germination and growth of plant pathogens [17,32,33]. Bordoh et al. reported that ginger crude extract at 10.0 mg/mL showed an effective antifungal effect against Colletotrichum gloeosporioides in vitro; it suppressed conidial germination and mycelial growth by 88.48% and 87.50%, respectively [17]. Numerous antimicrobial substances, such as antimicrobial peptides, organic acids, and plant essential oils that exert antifungal effects by increasing cell membrane permeability, damaging respiratory metabolism and energy metabolism, and reducing cell wall-degrading enzyme activities have been reported [24,34].

Zhou et al. and Peng et al. reported that the infection of *Fusarium solani* to ginger rhizome, but all ginger rhizomes were uniformly wounded with a sterilized borer to facilitate the infection. However, in the field condition, we found that in ginger seedlings inoculated with *F. solani*, the aboveground stems and leaves showed wilting symptoms, whereas the underground rhizomes remained symptomless (Figure S2). The morbidity of the aboveground parts reached 70.12%, whereas that of the underground parts was 0% (Table S2). These findings suggested that antifungal compounds exist in the ginger rhizome [12]. Therefore, in vitro experiments were performed to investigate the antifungal effects of GRE on *F. solani*. With increasing concentration from 1.25 to 20 mg/mL, GRE dose-dependently inhibited the colony growth and spore germination of *F. solani*, with the

strongest inhibitory effect at 20 mg/mL. Similarly, Mengal et al. reported that 15 mg/mL of ginger extract inhibited the mycelial growth of *Cladosporium cladosporioides* [35]. SEM analysis showed that 10 mg/mL GRE affected the mycelial morphology and structure of the pathogen.

Relative conductivity is an important indicator of cell membrane permeability [36,37]. Numerous antimicrobial substances have been shown to damage cell membrane integrity in pathogens, thus increasing the relative conductivity of mycelia [38]. In this study, GRE dose-dependently increased the relative conductivity of hyphae between 1–6 h of treatment. These results showed that GRE enhances the cell membrane permeability, leading to leakage of cell contents and inhibition of fungal growth.

Soluble proteins and sugars supply energy for the physiological metabolism and life activities of fungi [27,39]. In fungi and bacteria, cell membrane damage affects protein and sugar synthesis and utilization and thus, their contents [40]. Chen et al. demonstrated that 7-dimethoxytyrosine decreased the soluble protein and sugar contents in mycelia of *Penicillium italicum* due to cell leakage [41]. Similar results were obtained in this study: soluble protein and sugar contents in mycelia decreased gradually from day 1 to 5 after 10 mg/mL GRE treatment, which may be partly due to the changes in hyphal membrane permeability. It should be noted that the soluble protein and sugar contents in the control also slightly decreased with the prolongation of culture time (after 4 days of treatment), which may be due to the aging of the mycelia, leading to soluble protein and sugar leakage.

Most pathogenic microorganisms produce cell wall-degrading enzymes to eliminate plant anti-infection barriers and to uptake nutrients [9]. Pectinase and β-glucosidase are cell wall-degrading enzymes secreted by pathogenic fungi that play important roles in the invasion and initial expansion of pathogens [42]. In this study, GRE suppressed the activities of pectinase and β-glucosidase in *F. solani*. This result indicated that GRE reduced the ability of *F. solani* to decompose the host cell wall, decreasing its infection ability. Notably, the pectinase activity in control mycelia also slightly decreased from day 3 to 5, which may be due to the slowing down of metabolism with the prolonged culture time.

SDH and MDH are two important oxidoreductases in the tricarboxylic acid cycle and are key enzymes in energy metabolism [43]. Peng et al. found that crape myrtle extract suppressed the activities of SDH and MDH in *P. italicum*, indicating that the respiratory metabolic energy in the fungus was reduced [44]. Jiang et al. reported that forsythia inhibited the SDH and MDH activities in *Staphylococcus aureus*, thus inhibiting energy production [14]. Similarly, in this study, GRE treatment significantly decreased the activities of SDH and MDH, indicating that the energy metabolism of cells was strongly affected. With the attenuation of cellular respiration and metabolism and the insufficient energy supply, the infection ability of *F. solani* was decreased or even lost.

FA is a non-host-specific toxin. The FA production capacity is an important determinant of the pathogenicity of pathogens [45]. Generally, higher FA contents in pathogens are associated with stronger pathogenicity. In this study, GRE treatment reduced the pathogenicity of *F. solani* by inhibiting FA production, as the FA content decreased with prolonged treatment time. Chen et al. reported that *Trichoderma harzianum* SQR-T037 significantly decreased the FA content in *F. oxysporum* and could control fusarium wilt in cucumber [46]. As FA is a secondary metabolite, we hypothesize that the decrease in FA may be due to the disruption of the cell wall structure and respiratory metabolic pathways by GRE, which would have affected the synthesis of secondary metabolites.

To identify the antifungal components in GRE and unravel the antifungal mechanism of GRE, UPLC-MS/MS was performed. In total, 393 components belonging to eight categories were detected. Five compounds, 4-hydroxybenzaldehyde, quercetin, 6-Gingerol, scutellarein, and methyleugenol were selected for antifungal assays. Compounds such as gingerols are considered to be the most important medicinal compounds in ginger [47]. 6-Gingerol accounted for 1.05% of the rhizome content (Table S1), and 0.4mg/mL 6-Gingerol showed inhibitory effect on the growth of *F. solani* in synthetic medium. Similarly, methyleugenol (0.013%) (Table S3), an antifungal constituent in essential oil isolated

from many plant species, showed antifungal effect on *F. solani*. 4-Hydroxybenzaldehyde, a phenolic acid, accounted for 4.19% of the rhizome content (Table S1). It has been reported to exert antimicrobial activity against *Escherichia coli* and *Candida albicans* [48]. Quercetin, a flavonoid, accounted for 0.086% of the rhizome content (Table S1), and has been proven to have inhibitory activity against various pathogenic microorganisms [49]. In this study, 4-hydroxybenzaldehyde and quercetin displayed significant inhibitory activity against *F. solani* in synthetic medium, suggesting that they contribute to the antifungal activity of GRE against *F. solani*. Additionally, we found that the antifungal effect is not determined by the content of the component. For example, scutellarein, which accounted for 0.635% of the rhizome content, showed no inhibitory activity against *F. solani* (Table S3). Additionally, it should be noticed that the effect of the mixture showed better anti-microorganism activity than a single component [50,51]. Effective and safe single antimicrobial compounds and their mechanisms need to be studied in future as they may be suitable natural alternatives to chemical fungicides to protect crops and reduce post-harvest losses.

## 5. Conclusions

GRE exerts antifungal activity against *F. solani* by destroying the mycelia. 4-Hydroxybenzaldehyde and quercetin detected in GRE show antifungal effects on *F. solani*. This study indicated that that GRE and its phenolic and flavonoid constituents are excellent sources of antifungal compounds for the control of soilborne fungal diseases in the field as well as post-harvest crop diseases (Figure 8). Further research is needed to explore more components with antifungal activities in GRE and their antimicrobial mechanisms, which may provide foundation for the isolation and/or synthesis of these antibacterial substances that can be used as pesticides.

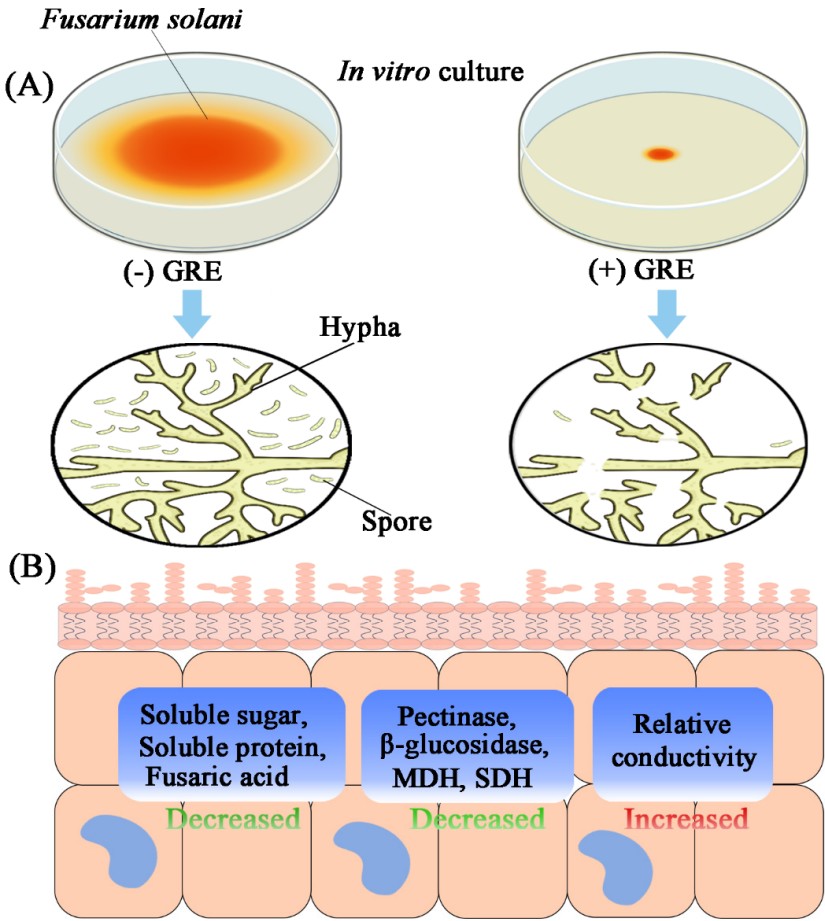

**Figure 8.** *Cont.*

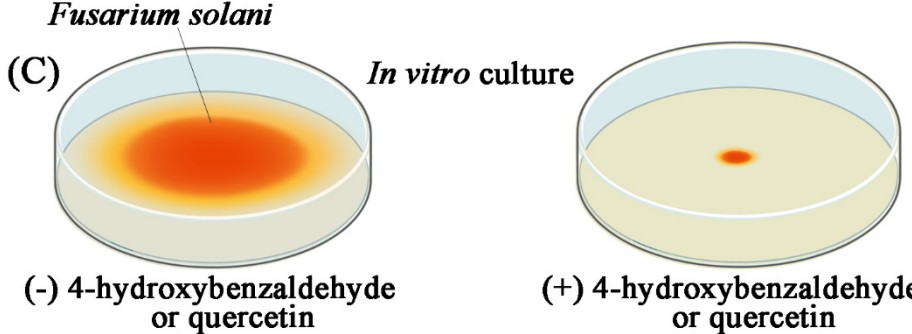

**Figure 8.** Mechanism diagram of GRE antifungal activity against *F. solani.* (**A**) In vitro antifungal activities of GRE against *F. solani.* Briefly, the colony diameter and spores' germination were significantly decreased, and hypha were ruptured. (**B**) The physiological changes of *F. solani* responded to GRE treatment. Generally, GRE decreased the cell membrane integrity of *F. solani,* which further increased relative conductivity of the fungal cell, and decreased soluble protein and sugar content. Meanwhile, the cell wall-degrading enzyme activities (pectinase and β-glucosidase), key enzymes in energy metabolism (MDH and SDH), and fusaric acid content were severely affected by GRE. (**C**) In vitro antifungal activities of 4-hydroxybenzaldehyde and quercetin against *F. solani.* Colony diameter was significantly decreased by 4-hydroxybenzaldehyde and quercetin.

**Supplementary Materials:** The following supporting information can be downloaded at: https://www.mdpi.com/article/10.3390/horticulturae8110983/s1, Figure S1: Morphologic changes of ginger aboveground (A) Control, treatment with distilled water; (B) Infection with F. solani for 30 days; and rhizome (C) Control, treatment with distilled water; (D) Infection with F. solani for 7 days; Figure S2: Effects of different concentrations of 6-gingerol on F. solani growth(A). Effects of different concentrations of scutellarin on F. solani growth(B). Effects of different concentrations of Eugenol methyl on F. solani growth(C); Table S1: Chemical components and their content in GRE and classification of them; Table S2: Morbidity of ginger plants and rhizomes inoculated by F. solani for 30 days and 7 days, respectively; and Table S3: The concentrations of 4-hydroxybenzaldehyde, quercetin, [6]-Gingerol, Scutellarein (5,6,7,4′-Tetrahydroxyflavone) and Methyleugenol in the rhizome and aboveground.

**Author Contributions:** Conception and designing, Y.-Q.L.; Conducting experiments and writing—original draft preparation, J.Z., G.L., K.-Y.X. and S.-J.X.; Writing—review and editing, Y.-X.Z. and J.-L.Y.; Supervision, Y.-X.Z.; Data analysis, J.-W.M. and C.-Q.G. All authors have read and agreed to the published version of the manuscript.

**Funding:** This study was supported by the Key Research and Development program of Hubei province, China, No. 2022BBA0061; the National Natural Science Foundation of Hubei Province, No. 2021CBF512. This research was funded by Scientific Research Program of Hubei Provincial Department of Education (No. D20201301).

**Institutional Review Board Statement:** Not applicable.

**Informed Consent Statement:** Not applicable.

**Data Availability Statement:** Not applicable.

**Conflicts of Interest:** The authors declare no conflict of interest.

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
