# Peer review of "Antifungal Activity of Ginger Rhizome Extract against Fusarium solani"

_horticulturae, doi:10.3390/horticulturae8110983_

Round 1

Reviewer 1 Report

Summary. The authors report on the in vitro antifungal effect of ginger Rhizome extract (GRE) on Fusarium solani. They present a chemical profile of the extract and choose 4-hydroxybenzaldehyde (HBA) and quercetin as two possible candidates for the fungicidal effect. Beside the inhibitory effects of GRE and the chemical on agar plates or SEM of the mycelium they show the effects on cell membrane, proteins, sugars, cell wall degrading enzymes, the respiratory metabolism, and fusaric acid. In the supplement they also show an infection of F. solani on ginger.

You report on GRE as a new alternatrive for synthgetic fungicide and agricultural practice for sure needs as many alternatives as possible to protect crops against diseases. In this context your study is reporting on a "new" alternative against Fusarium solani.

That plant extracts could have antimicrobial effects is not new, especially if they contain a lot of phenolic acid derivatives, essentil oils (to mention only two groups). Your hypothesis to use GRE is based on the observation that F. solani is not able to infect ginger rhizomes (L56-62). In this context you did not mention two studies which show that F. solani could infect rhizomes. So the hypothesis is probably based on an observation with a strain, who is (for whatever reason)  apathogenic to rhizomes. You do not report on details of the used strain. So an evaluation is not possible. But you could have used Fusarium strains available at the Spice Crops Research Institute. It is unclear why you did not use themm (ee also commenbt below).

It is unclear how much rhizomes (weight) you have extracted to produce the GRE. This would be important for practical reasons. Would it be possible to produce enough GRE for an application in the field. You need to give an estimate, if one could produce enough GRE.

You only report on IN VITRO data. There are no data on the fungistatic / fungicidal efficacy of GRE on plants. You have used 30% alcohol, which would not be possible in a spray application. Data on dissolving GRE in water are missing. Youu need to address this issue in your discussion.

To use HBA and Q is based on their content in rhizomes (in contrast to shoot) and that antimicrobial effects are reported. On the first criteria: what is about the other phenolic acids or alcaloids, which occur in higher concentrations. Especially the concentration o quercetin is very low in rhizomes and it is questionable, if this substance would make an effect in the field. On the 2nd criteria: there are only one report for HBA and quercetin and the reported effects in [46, 47] are not impressing. You should give a more detailed description why you have chosen these two compounds and none of the other ones.

Beside this more general concerns I have more specific concerns which I list in the following. More detailed comments could be found in the attached PDF of your manuscript.

Work with literature / citations. Please do not use secondary or even tertiary citations and references in introduction. I have given examples in the attached PDF. It is good scientific practice that the reader will find the information in the cited article (1st order citation). The authors you are referring to must have produced the data on their own! I have checked the literature in your introduction and found several examples of wrong citations, non citing available information or not suited references. As a reader I expect that the authors and co-authors check carefully the references. This is madatory (!) and requires a major careful revision of the manuscript.

It is mandatory to cite / mention articles from co-athors, if the studies have a direct relation to the study. I found two studiey where Jie Zhou on F. solani on ginger (1st author and co-author in the article by Peng et al),  but you do not mention it concerning pathogenicity of rhizomes in your introduction. It is of importance, because this study shows that F. solani could infect rhizomes. You MUST address these different results properly in your manuscript!

Pathogenicity of F. solani. You report that in a preliminary study F. solani was not able to infect rhizomes and concluded that rhizomes contain antifungal compound (L56-58). It raises the question about the pathogenicity of your strain. Peng et al [your ref. 12] showed infection of F. solani on rhizomes (but you do not mention it). More important, Jie Zhou (co-author of the manuscript) published recently an article showing infection of rhizomes too.  You need to address this evidence from literature in your manuscript, because it is the base of your hypothesis. You need to explain where your strain was coming from or why you did not use the strain used in the article by Zhou. You shoukld rule out the possibility that your strain lose pathogenicity by passaging over plants to increase pathogenicity.

SEM results. It is not clear how many pictures you have taken and if the shown photos represent the treatment. In Fig 2B is mycelium shown for 10 GRE. According to Fig 1 there is hardly not outgrowth of the fungus to be seen. I am wondering if the effect you see in figure 2 is based on the age of the mycelium. While you have new growing mycelium in control, you might have collected mycelium from GRE which has alraedy been stopped in growth. Hod did you check, that you have sampled mycelium of the same physiological age and not mycelium undergoing senescence already. In this case it would help to show how "old" mycelium on the control would look like.

F. solani inoculation assays. They are describe in material and methods (L88-105) but results are missing in the manuscript but are only available in S2! Are these results required? You only show that shoots are getting infected and ginger rhizomes not. This would have no crucial impact on the use of GRE as antifungal extract. In S2 the reported morbidity of 2/3 is not reflected by the picture. One could see some wilting but sure no morbidity. Did you measure a wilting index? Must be revised.

Missing in vivo assay. You did not perform an in vivo assay where you treated plants with GRE as a antifungal extract against Fusarium solani.. This is important here, because you have used an extract in 30% alcohol. It might not have an effect in vitro, but sure such an alcohol concentration would show severe phytotoxic effects when you need to spray it. Hence, an alcohol exctract is maybe not suited for application of GRE on plants. But you did not show, if GRE dissolved in water could achieve the same effects in vico as shown in vitro.

On the inhibitory effect of 4-hydroxybenzaldehyde. The maximum amount of GRE was 20mg/ml. This GRE contains 4.18% of 4-hydroxybenzaldehyde. This would result in a concentration of 4-hydroxybenzaldehyde of about 0.8 mg /ml. In your experiment the lowest concentration was 2mg/ml. How could you explain it? The same could be said about quercitin, which concentration in GRE is much lower.

Concluding Remarks. You should clearly address all concerns raised above. What I think is impiortant that you discuss the meaning of your findings for the practice. In this context you should point out that this was an in vitro study and I suggest to change the title accordingly. But the data delivered in the manuscript a worthy of publication (especially the detailed results on the different effects of GRE on Fusarium). But the manuscript must be revised to meet the raised concerns here and in the PDF.

Reviewer 2 Report

The paper is interesting, the research is well conducted and the results are sound.

I have only found some minor points.

One has to do with typography (lettering too small), but in figure 7 the P values has not been Nowadays there is a lot of concern about the use of chemical pesticides, so the search for novel formulations based on natural products is a hot topic in current horticulture. Ginger is a standard remedy in traditional chines medicine, and also is widely used in Europe for its medicinal properties, among those properties the Authors evaluate its fungicide effect. This effect has been assessed both by the evaluation of growth (fig 1) and also by scanning electronic microscopy (fig 2). This double check is not easy to find in the literature. The other figures evaluate the effects on different parameters. Altogether the evidence is very convincing. The only thing that could enrich the manuscript would be including in the discussion some information on the commercial perspectives of ginger extracts. Is very expensive? How much would it cost its use per Ha in open field or in a greenhouse? The extraction is scalable?
